# Nitrate Utilization Promotes Systemic Infection of *Salmonella* Typhimurium in Mice

**DOI:** 10.3390/ijms23137220

**Published:** 2022-06-29

**Authors:** Wanwu Li, Linxing Li, Xiaolin Yan, Pan Wu, Tianli Zhang, Yu Fan, Shuai Ma, Xinyue Wang, Lingyan Jiang

**Affiliations:** 1The Key Laboratory of Molecular Microbiology and Technology, Ministry of Education, Nankai University, Tianjin 300457, China; wanwuli@mail.nankai.edu.cn (W.L.); lilinxing19@mail.nankai.edu.cn (L.L.); 2120211278@mail.nankai.edu.cn (X.Y.); wupan2016@mail.nankai.edu.cn (P.W.); 1120180080@mail.nankai.edu.cn (T.Z.); 1120170077@mail.nankai.edu.cn (Y.F.); 2120191113@mail.nankai.edu.cn (S.M.); wangxinyue1120200566@mail.nankai.edu.cn (X.W.); 2Tianjin Key Laboratory of Microbial Functional Genomics, TEDA Institute of Biological Sciences and Biotechnology, Nankai University, Tianjin 300457, China

**Keywords:** *Salmonella* typhimurium, nitrate utilization, systemic infection, macrophages, SPI-2 genes

## Abstract

*Salmonella* Typhimurium is an invasive enteric pathogen that causes gastroenteritis in humans and life-threatening systemic infections in mice. During infection of the intestine, *S.* Typhimurium can exploit nitrate as an electron acceptor to enhance its growth. However, the roles of nitrate on *S.* Typhimurium systemic infection are unknown. In this study, nitrate levels were found to be significantly increased in the liver and spleen of mice systemically infected by *S.* Typhimurium. Mutations in genes encoding nitrate transmembrane transporter (*narK*) or nitrate-producing flavohemoprotein (*hmpA*) decreased the replication of *S.* Typhimurium in macrophages and reduced systemic infection in vivo, suggesting that nitrate utilization promotes *S.* Typhimurium systemic virulence. Moreover, nitrate utilization contributes to the acidification of the *S.* Typhimurium cytoplasm, which can sustain the virulence of *S.* Typhimurium by increasing the transcription of virulence genes encoding on *Salmonella* pathogenicity island 2 (SPI-2). Furthermore, the growth advantage of *S.* Typhimurium conferred by nitrate utilization occurred only under low-oxygen conditions, and the nitrate utilization was activated by both the global regulator Fnr and the nitrate-sensing two-component system NarX-NarL. Collectively, this study revealed a novel mechanism adopted by *Salmonella* to interact with its host and increase its virulence.

## 1. Introduction

*Salmonella enterica* serovar Typhimurium (*S.* Typhimurium) is a Gram-negative intracellular pathogen that typically causes mild self-limiting gastroenteritis in humans; however, mouse infection always results in life-threatening systemic infection with disease symptoms that resemble typhoid fever in humans [1,2]. To establish a systemic infection, *S.* Typhimurium must survive and replicate within mouse macrophages [3]. The liver and spleen, both of which contain large numbers of tissue-resident macrophages and recruit monocytes to give rise to monocyte-derived macrophages upon infection [4,5], are the primary tissues that colonize a high density of *S.* Typhimurium during systemic infection [6]. The virulence genes encoded in *Salmonella* pathogenicity island 2 (SPI-2) are essential for *S.* Typhimurium replication within mouse macrophages and for growth in systemic tissues [3]. The expression of SPI-2 virulence factors is required for the establishment and maintenance of *Salmonella*-containing vacuoles (SCV) that function as modified phagosomes in which *Salmonella* resides and replicates [7].

Nitrate is one of the most abundant metabolites produced during *S.* Typhimurium gut infection, and it is generated as a result of increased production of nitric oxide (NO), an antibacterial molecule that is actively produced by host cells [8]. It has been reported that *S.* Typhimurium infection stimulates the expression of inducible NO synthase (iNOS) encoded by *Nos2* in the cecal mucosa of mice, thus leading to an increase in the production of chemically active NO [8,9]. The diffusion-controlled reaction of NO with superoxide radicals forms short-lived peroxynitrate that is further converted to relatively stable nitrate [10]. Thus, inflamed intestines exhibit an increase in the production of host-derived nitrate. Additionally, host-derived nitrate has been demonstrated to play a central role in promoting *S.* Typhimurium fitness in the gut, and *S.* Typhimurium can exploit nitrate as an electron acceptor to enhance its growth in the inflamed intestine [9]. On the other hand, the host-derived nitrate is used by *S.* Typhimurium as an environmental cue to activate the expression of flagella [11], and this is important for *S.* Typhimurium adhesion and invasion of intestinal epithelial cells [12]. Although it is established that *S.* Typhimurium induces the production of NO and nitrate in the macrophages [11], the changes in nitrate levels in vivo during *S.* Typhimurium systemic infection remain unclear. Additionally, the functions of nitrate in intracellular replication and systemic infection by *S.* Typhimurium are unknown.

The expression of SPI-2 genes is induced by many environmental cues within macrophages, and acidic pH represents one of the most important cues [13,14]. It is well recognized that the environment of the macrophage SCV is acidic with an estimated pH range of <5 to 5.5 [15,16], and the acidic conditions of the SCV result in a decrease in cytoplasmic pH of *S.* Typhimurium [17]. Previous studies have reported that a decrease in cytoplasmic pH is required for the high expression of SPI-2 genes and for *S.* Typhimurium virulence [13,14]. The master virulence regulatory systems, including PhoQ/PhoP and EnvZ/OmpR, are implicated in the activation of SPI-2 genes via sensing the acidic cytoplasmic pH of *S.* Typhimurium [17,18]. The mechanisms related to the acidification of SCV and the *S.* Typhimurium cytoplasm remain poorly characterized. The recognition of *S.* Typhimurium by the Toll-like receptors (TLRs) of macrophages can enhance the acidification rate of SCV [19,20]; however, other pathways correlated with the acidification of SCV and bacterial cytoplasm remain unclear. Considering that the nitrate transmembrane transporter NarK is possibly a nitrate/proton symtransporter, and that nitrate utilization may produce protons [21,22,23], it is possible that nitrate transport and utilization contribute to the cytoplasmic acidification in *S.* Typhimurium.

This study aims to reveal the possible functions of nitrate in the systemic infection of *S.* Typhimurium and to illustrate the underlying regulatory mechanisms. The results demonstrate that the utilization of host-derived nitrate enhances the intracellular replication and systemic pathogenicity of *S.* Typhimurium. Nitrate utilization, which is activated by both the global regulator Fnr and the nitrate-sensing two-component system NarX-NarL in *S.* Typhimurium, promotes *S.* Typhimurium virulence via increasing the acidification of *S.* Typhimurium cytoplasm to trigger SPI-2 virulence gene expression. 

## 2. Results

### 2.1. Nitrate Levels Are Increased during Systemic Infection by S. Typhimurium

During systemic infection, *S.* Typhimurium resides predominantly within the host macrophages [24]. Previous studies have reported that *S.* Typhimurium infection can stimulate iNOS and nitrate production in mouse macrophages [11], and the increase in nitrate levels in *S.* Typhimurium-infected RAW264.7 macrophages was also observed in this study. Concentrations of nitrate as high as 20 μM were detected in the supernatant of RAW264.7 cells during the 24 h infection period with *S.* Typhimurium (Figure 1a), and this is consistent with the data reported in previous studies.

The changes in nitrate levels in vivo during *S.* Typhimurium systemic infection remain unclear. To understand whether nitrate levels are increased in the systemic tissues (liver and spleen) of *S.* Typhimurium-infected mice, BALB/c and C57BL/6 mice were infected with ~10^4^ cfu of *S.* Typhimurium by intraperitoneal injection (i.p.), and then the nitrate concentrations in mouse livers and spleens on day 5 post-infection were tested. The results revealed that nitrate levels were significantly increased in the liver and spleen of *S.* Typhimurium-infected BALB/c and C57BL/6 mice compared to those of the uninfected mice (Figure 1b,c). As nitrate can be reduced to nitrite, the nitrite concentrations in the mouse liver and spleen were also measured. However, nitrite levels were not significantly altered in the livers and spleens of *S.* Typhimurium-infected BALB/c mice (Appendix A). These results indicate that nitrate levels increased during the systemic infection of *S.* Typhimurium both in vitro (macrophages) and in vivo (mouse liver and spleen).

We also investigated whether the increase in nitrate of *S.* Typhimurium-infected mouse liver and spleen is derived from the host by using iNOS (inducible nitric oxide synthase)-deficient mice (iNOS^−/−^ mice harboring a mutation in the *Nos2* gene). It is established that the generation of nitrate within the host is primarily dependent upon iNOS that is encoded by the *Nos2* gene and catalyzes L-arginine to NO [8,9]. *S.* Typhimurium infection did not increase nitrate concentration in the livers and spleens of iNOS^−/−^ mice (Figure 1d), thus implying that the increase in nitrate in the systemic tissues of *S.* Typhimurium-infected wild-type mice was derived from the host.

Taken together, these results indicate that nitrate levels are increased during systemic infection with *S.* Typhimurium and that the increased nitrate is primarily derived from the host.

### 2.2. Nitrate Utilization Promotes S. Typhimurium Replication in Macrophages

As *S.* Typhimurium can exploit nitrate as an electron acceptor to enhance its growth in the intestine [9], whether *S.* Typhimurium can utilize nitrate to promote its systemic infection was evaluated. It is established that *S.* Typhimurium can utilize two potential routes to gain nitrate from the host, including (1) direct uptake of nitrate from the environment via its nitrate transmembrane transporters NarK and NarU that are encoded by *narK* and *narU* gene, respectively [25]; and (2) synthesis of nitrate from NO and O_2_ by a flavohemoprotein (encoded by *hmpA* gene) within the bacterial cells based on the ability of host-derived NO to freely diffuse into the SCV [26]. To test the influence of nitrate utilization on the systemic virulence of *S.* Typhimurium, mutants lacking *narK* (Δ*narK*), *narU* (Δ*narU)*, and *hmpA* (Δ*hmpA*) were generated. The mutant strains grew, as well as the wild-type (WT) strain, in LB medium and DMEM (Figure 2a,b, respectively), thus indicating that the mutation of these genes did not influence *S.* Typhimurium growth in vitro.

As replication within host macrophages is essential for *S.* Typhimurium systemic infection, whether the utilization of nitrate influences bacterial replication in macrophages was tested via macrophage replication assays. Nitrate supplementation in DMEM significantly increased the replication of *S.* Typhimurium in RAW264.7 cells, thus indicating that nitrate plays a potential role in promoting *S.* Typhimurium replication within macrophages (Figure 2c). The mutation of *narK* and *hmpA* significantly decreased the replication of *S.* Typhimurium in RAW264.7 cells (Figure 2d), while the mutation of *narU* did not significantly influence the replication ability of *S.* Typhimurium (Appendix A). A previous study revealed that NarK plays a central role in nitrate uptake across the membrane in diverse bacteria [27], while NarU always provides a bacterial survival advantage during severe nutrient starvation or slow growth [28]. The results indicate that nitrate utilization contributes to *S.* Typhimurium replication within macrophages and that the uptake of host nitrate is dependent upon NarK but not upon NarU. Moreover, *hmpA* and *narK* double mutant strain (Δ*hmpA*Δ*narK*) was generated, and the replication of Δ*hmpA*Δ*narK* was also significantly decreased compared to that of the WT, thus further confirming the importance of nitrate utilization in *S.* Typhimurium intracellular replication (Figure 2d).

### 2.3. Nitrate Utilization Promotes S. Typhimurium Systemic Infection in the Mice

We next tested whether the utilization of nitrate affected *S.* Typhimurium systemic virulence in vivo. Mouse infection assays revealed that supplementation with nitrate sodium in drinking water provided a fitness advantage for *S.* Typhimurium to colonize the mouse liver and spleen (Figure 3a), thus indicating that nitrate plays a potential role in promoting *S.* Typhimurium systemic virulence in vivo. Mutations in *narK* and *hmpA* significantly increased the survival rate of the infected mice (Figure 3b), and the bacterial burdens in the livers and spleens of Δ*narK*-infected and Δ*hmpA*-infected mice were significantly decreased compared to those of the wild-type (WT)-infected mice (Figure 3c), thus verifying that nitrate utilization contributes to *S.* Typhimurium systemic infection. Consistent with the results of macrophage replication assays, both the survival rate and bacterial burden in the livers and spleens of Δ*narU*-infected mice were comparable to those of WT-infected mice (Appendix A). However, Δ*hmpA*Δ*narK* was more attenuated than were the Δ*narK or* Δ*hmpA* single mutants (Figure 3b,c), thus revealing that both nitrate uptake and nitrate synthesis are important for *S.* Typhimurium systemic virulence in vivo. Collectively, these results indicate that nitrate utilization promotes *S.* Typhimurium systemic infection in mice.

### 2.4. Nitrate Utilization Increases the Acidification of the S. Typhimurium Cytoplasm to Activate Virulence Gene Expression

Next, the underlying mechanisms related to the ability of nitrate utilization to promote *S.* Typhimurium systemic infection were investigated. Because SPI-2 genes are the virulence genes that are required for *S.* Typhimurium intracellular replication and systemic infection [7], whether the mutation of *narK* influences the expression of SPI-2 genes was investigated. qRT-PCR analysis showed that the mutation of *narK* significantly decreased the mRNA levels of five representative SPI-2 genes (*ssaG*, *ssaH*, *sifA*, *sifB*, and *sseL*) (Figure 4a,b), indicating that nitrate utilization plays a role in activating SPI-2 gene expression. However, it remains unclear how nitrate utilization activates the SPI-2 genes.

Interestingly, the utilization of nitrate was found to be associated with the acidification of the *S.* Typhimurium cytoplasm, and acidic pH is a well-characterized signal that activates SPI-2 gene expression. When there was no nitrate in the medium, the pH of the medium cultured with the WT strain was not significantly different from that of the Δ*narK* mutant (Figure 4c). However, when nitrate was added to the medium, the pH of the medium cultured with the WT strain was significantly lower than that cultured with the Δ*narK* mutant (Figure 4c). Moreover, the decreased pH levels of the medium in which the WT strain was cultured were more obvious with an increase in nitrate concentration. However, this decrease was not significant in the Δ*narK* mutant (Figure 4d). These results indicate that the utilization of nitrate by *S.* Typhimurium may generate protons that can be released into the extracellular environment, thus resulting in acidification of the medium. BCECF-AM is a commonly used fluorescent probe that detects intracellular pH, and the intrabacterial pH of *S.* Typhimurium was detected by using BCECF-AM [13,17]. The intracellular pH was significantly decreased with an increase in nitrate concentration in the medium; however, this decrease was not significant for the Δ*narK* mutant (Figure 4e), thus indicating that the utilization of nitrate by *S.* Typhimurium resulted in cytoplasm acidification of the bacteria. Together, these results reveal that nitrate utilization increases the acidification of the *S.* Typhimurium cytoplasm to activate SPI-2 virulence gene expression, thus promoting *S.* Typhimurium systemic infection.

### 2.5. Nitrate Utilization Promotes S. Typhimurium Growth under Low-Oxygen Conditions

We next investigated the ranges of nitrate levels that can promote *S.* Typhimurium growth and also the conditions under which the utilization of nitrate can promote *S.* Typhimurium growth. The growth promotion index (GPI) was used to objectively represent the effect of nitrate on the growth of *S.* Typhimurium, and GPI was calculated as the OD_600_ of the bacterial culture supplemented with nitrate divided by that without nitrate under the same culture conditions. With increasing nitrate concentration in N-minimum medium (from 0 to 15 mM), the OD_600_ of the *S.* Typhimurium culture was significantly increased as reflected by the increased GPI (Figure 5a). However, a continuous increase in the nitrate concentration (30 to 45 mM) in the medium did not improve *S.* Typhimurium growth compared to that at 15 mM (Figure 5a), thus suggesting that excess nitrate cannot be effectively utilized. Despite the high concentration of nitrate that existed and could not be consumed, it did not significantly inhibit the growth of *S.* Typhimurium (Figure 5a), thus indicating the strong resistance of *S.* Typhimurium to nitrate. These results indicate that nitrate promotes *S.* Typhimurium growth in vitro within a wide range (from 0 to 45 mM) as indicated by our experimental data.

Considering that the oxygen concentration within macrophages is relatively low and nitrate utilization promotes *S.* Typhimurium growth in macrophages, we tested whether nitrate-induced *S.* Typhimurium growth is related to oxygen concentration. With increasing oxygen levels (oxygen volume/medium volume, OV/MV), the OD_600_ of *S.* Typhimurium became increased regardless of the presence of nitrate (Table 1), thus indicating that oxygen is an important factor that promotes bacterial growth. However, the addition of 30 mM nitrate increased *S.* Typhimurium growth when the OVP/MV was in the range of 0–0.4, but this was not observed at the OVP/MV of 1 (Table 1). GPI decreased with increasing oxygen levels (Table 1 and Figure 5b). These results revealed that oxygen impaired the promotion effect of nitrate on *S.* Typhimurium growth. The addition of 30 mM nitrate significantly increased the growth of *S.* Typhimurium under low-oxygen conditions but not under high-oxygen conditions (Figure 5c). Additionally, the promotion of nitrate to *S.* Typhimurium growth was independent of carbon sources, as using glucose, pyruvate, or lactate as the sole carbon source did not influence nitrate-induced *S.* Typhimurium growth (Figure 5c).

Together, these results indicate that nitrate promotes *S.* Typhimurium growth under low-oxygen conditions, which are just the environmental cues inside macrophages.

### 2.6. Nitrate Utilization Is Activated by Both Fnr and the Nitrate-Sensing Two-Component System NarX-NarL under Low-Oxygen Conditions

The regulatory system associated with nitrate utilization was also investigated. Previous studies reported that the transcriptions of nitrate utilization genes in *E. coli*, including the genes encoding nitrate transporter (*narK*) and nitrate reductase component (*narG* and *napF*), are all activated by the global regulator Fnr that functions under oxygen-limiting conditions [29,30]. Given that nitrate utilization promotes *S.* Typhimurium growth under low-oxygen conditions, it is likely that Fnr also activates nitrate utilization genes in *S.* Typhimurium. Under low-oxygen conditions, the addition of 30 mM nitrate into N-minimal medium significantly increased the transcription of *narK*, *narG*, and *napF* in the WT strain, thus suggesting the activating role of nitrate on these genes (Figure 6a). Additionally, the mutation of *fnr* abolished the activation effect of nitrate on these genes (Figure 6a), thus revealing that Fnr activates nitrate utilization in response to low-oxygen conditions.

Nitrate can also activate the nitrate-sensing two-component system NarX-NarL in *E. coli*, with NarX being the sensor that responds to nitrate and NarL being the regulator that can induce the transcription of *narK, narG*, and other genes [31,32]. Next, whether NarX and NarL are involved in the regulation of nitrate utilization in *S.* Typhimurium was investigated. The mutation of *narX* or *narL* significantly decreased the transcription of *narK*, *narG*, and *napF* under low-oxygen conditions (Figure 6b), thus confirming the positive regulation of nitrate utilization genes by NarX-NarL in *S.* Typhimurium. EMSA experiments showed that NarL could bind to the promoter regions of *narK* and *narG* to directly activate the expression of the two genes (Figure 6c).

Consistent with the above results indicating that nitrate utilization plays a role in activating SPI-2 gene expression, the mutation of *fnr* or *narL* significantly decreased the transcription of the five representative SPI-2 genes (Appendix A).

Together, these results indicate that nitrate utilization in *S.* Typhimurium is activated by both Fnr and the nitrate-sensing two-component system NarX-NarL: Fnr is responsive to low-oxygen conditions within the host cells to activate nitrate utilization genes, and NarX-NarL responds to the increased nitrate concentration within host cells to directly activate nitrate utilization genes.

## 3. Discussion

*S.* Typhimurium primarily resides within macrophages during systemic infection [33,34]. As critical effector cells of the innate immune system, macrophages possess various antibacterial mechanisms and are activated during infection with *S.* Typhimurium. However, *S.* Typhimurium has evolved numerous strategies to resist killing by host defense mechanisms and can even exploit these mechanisms for their own purposes [35]. NO and reactive nitrogen species (RNS) generated by activated macrophages are important antimicrobial effectors [36], and *S.* Typhimurium is known to avoid the entry of RNS into SCV by expressing SPI-2 effector proteins [36,37]. As a stable derivative of NO, the levels of host-derived nitrate always increase due to the generation of NO upon *S.* Typhimurium infection. During infection of the intestine, *S.* Typhimurium can exploit this increased host nitrate to promote intestinal pathogenicity [11]. However, the functions of nitrate in the replication of *S.* Typhimurium within macrophages and in the systemic infection of *S.* Typhimurium are unknown. Our results clearly demonstrate that *S.* Typhimurium can also use nitrate to promote its intracellular replication and systemic infection, thus implying that *S.* Typhimurium can not only avoid NO itself within macrophages but can also use the downstream products of NO to promote its own fitness. Our results highlight that *S.* Typhimurium has evolved sophisticated mechanisms to evade the host immune response or exploit host functions to facilitate its infection process.

*S.* Typhimurium employs nitrate as an electronic receptor to promote its growth in the intestinal tract [8,9], and it exploits nitrate as a signal to stimulate the expression of flagella to promote bacterial adhesion and invasion of intestinal epithelial cells [11] to thereby activate its intestinal pathogenicity. Our results indicate that nitrate also activates *S.* Typhimurium SPI-2 gene expression by inducing cytoplasmic acidification during systemic infection. From these results, we can conclude that *S.* Typhimurium uses nitrate in different ways during different infection stages, primarily due to the observation that the virulence genes required by the pathogen are different in intestinal and systemic infections. It is established that environmental signals are critical factors that control the expression of virulence genes [11,38,39]. Although flagellar genes are activated during intestinal infection, they must be repressed during systemic infection due to the strong immunogenicity of flagellar proteins [38]. Various mechanisms related to the inhibition of flagellar genes within macrophages have been characterized, including acid signaling that can inhibit flagellar genes expression via repressing the regulatory protein AsiR as reported in our previous study [38,39]. Therefore, although nitrate may also possess the potential to stimulate the expression of flagellar genes in macrophages, flagellar genes are inhibited by various environmental cues in macrophages and regulatory systems. On the other hand, nitrate utilization can promote cytoplasmic acidification of *S.* Typhimurium, and the acidification-induced inhibition of flagellar gene expression during nitrate utilization also counteracted nitrate-mediated flagellar gene activation.

As acidic pH is a crucial signal to activate SPI-2 gene expression, acidification of the surrounding environment and the cytoplasm of *S.* Typhimurium both play an essential role during *S.* Typhimurium systemic infection [13,14]. The TLR-mediated recognition of bacteria is known to promote SCV acidification [20,40]; however, other mechanisms that mediate the acidification of the SCV and bacterial cytoplasm are largely unknown. Our results report for the first time a mechanism for the cytoplasmic acidification of *S.* Typhimurium via the transportation and utilization of nitrate. Additionally, TLRs mediate several immune signaling pathways, including the NF-κB and STAT pathways that activate the downstream expression of iNOS [41] to stimulate the production of NO and nitrate. It has been found that effector protein SopE2 of *S.* Typhimurium enhances the production of iNOS and nitrate in a colitis model, and SopE2 is one of the main factors responsible for the activation of NF-κB pathways during *Salmonella* infection, in addition to other effector proteins and bacterial LPS [9]. In this study, the utilization of nitrate generated downstream of NF-κB pathways just promotes acidification of the *S.* Typhimurium cytoplasm. Therefore, it is possible that nitrate utilization establishes a link between SCV acidification and bacterial cytoplasmic acidification.

We demonstrate that nitrate utilization stimulates *S*. Typhimurium growth only under low-oxygen conditions, and these are cues within macrophages. Thus, the activation of SPI-2 gene expression mediated by nitrate utilization occurs within host macrophages but does not occur during in vitro growth. Our results also indicate that nitrate utilization is activated by both the hypoxia regulator Fnr and the nitrate-sensing two-component system NarX-NarL. The use of the two regulatory systems to activate the nitrate utilization pathway may collectively ensure the high expression of SPI-2 genes. At the early stage of infection, the concentrations of NO and nitrate within macrophages are relatively low, and nitrate utilization is activated by Fnr sensing of low-oxygen cues that then enhances the cytoplasmic acidification of *S*. Typhimurium to activate SPI-2 genes. NO and nitrate concentrations are increased along with the infection process, and NarX-NarL directly senses the high nitrate concentration in the surrounding environment to activate the nitrate utilization pathway that further enhances cytoplasmic acidification and SPI-2 gene expression.

Overall, our results reveal the importance of nitrate utilization for *S*. Typhimurium systemic infection and uncover the mechanisms associated with how nitrate utilization promotes *S*. Typhimurium systemic infection and the regulatory networks of nitrate utilization in *S*. Typhimurium within macrophages. Furthermore, our findings emphasize the critical role of the SCV and cytoplasmic acidification in the activation of SPI-2 genes. In addition to OmpR and PhoP that were previously demonstrated to activate SPI-2 genes via sensing the acidification of *S*. Typhimurium cytoplasm [17,18], Fnr and NarX-narL have also been demonstrated to promote SPI-2 gene expression by activating nitrate utilization to enhance cytoplasmic acidification in this study. However, the cross-talk between Fnr and OmpR- or PhoP-associated regulatory pathways requires further investigation.

## 4. Materials and Methods

### 4.1. Bacterial Strains and Plasmids

*Salmonella* Typhimurium ATCC 14028s was used as the wild-type (WT) strain in this study. The mutant strain was generated using the λ Red recombinase system supported by the pSim17 plasmid (blastidin-resistant) encoding three proteins (Exo, Beta, and Gam) that are required for homologous recombination [42]. First, a WT strain harboring the pSim17 plasmid was established through electrotransformation. Then, DNA fragments composed sequentially (5′→3′) of an upstream 50–70 bp sequence of the target gene, the chloramphenicol resistance gene sequence, and the downstream 50–70 bp reverse complementary sequence of the target gene were amplified by PCR. The chloramphenicol-resistant pKD3 plasmid was used as the template of PCR, and primers were designed to cover the 50–70 bp homologous arm sequences. Next, the DNA fragments were introduced into competent bacterial cells by electrotransformation. Finally, the cells were cultured at 37 °C for 2 h, and the suspensions of recovered bacteria were spread onto agar plates containing 25 μg/mL of chloramphenicol to obtain single mutant colonies. To validate the correctness of the mutation, the target mutation loci of the chloramphenicol-resistant single colony were PCR-amplified and preliminarily identified by electrophoresis. Further identification was performed using Sanger sequencing. The Δ*hmpA*Δ*narK* double mutant was established based on a chloramphenicol-resistant Δ*hmpA* mutant by introducing a kanamycin-resistance gene (from pKD4 plasmid) to replace the *narK* gene.

To express and purify the NarL protein with 6×His-tag, the *narX*-*narL* fusion gene was cloned into the pET-28 a (+) plasmid between the NcoI and XhoI sites, and the constructed plasmids were transformed into *Escherichia coli* BL21 (DE3). All constructed plasmids and strains were validated using the methods described for identification of the mutant strains. The primers used in this study are listed in Appendix A.

### 4.2. Conditions for the Culture of Bacteria and RAW 264.7 Macrophages

The bacteria are generally cultured in LB medium (10 g/L of tryptone, 5 g/L of yeast extract, and 10 g/L of NaCl) at 37 °C if no clear explanation is stated. When necessary, antibiotics were added to the medium individually or together at the following final concentrations: 25 μg/mL of chloramphenicol; 50 μg/mL of kanamycin; 200 µg/mL of blasticidin.

RAW 264.7 macrophages used in this study were cultured in Dulbecco’s modified Eagle’s medium (DMEM) with 10% fetal bovine serum at 37 °C under 5% CO_2_. At 24 h prior to infection with bacteria, RAW 264.7 macrophages were scraped off and aspirated to obtain a homogeneous cell suspension, and the cells were seeded into 12-well culture plates to achieve final cell monolayers.

### 4.3. Evaluating the Virulence of S. Typhimurium in Mice

Animal studies were conducted according to protocols approved by the Institutional Animal Care Committee of Nankai University (Tianjin, China). BALB/c mice and C57BL/6 (female, 7 weeks old, ~25 g) were purchased from Beijing Vital River Laboratory Animal Technology Co., Ltd. (Beijing, China). iNOS^−/−^ mice with a C57BL/6 background were purchased from Jackson Laboratory (USA). All mice were reared in a ventilated cage with a rack that provided free access to chow and water. The temperature of the room where the mice were housed was approximately 25 °C and included a natural light cycle. To determine the survival rate, overnight cultures of *S.* Typhimurium were diluted to 1 × 10^6^ CFU/mL with normal saline (0.9% NaCl solution), and 100 µL of the diluted bacterial suspension was injected into the right peritoneum. The number of dead mice was recorded daily, and survival rates were calculated. To calculate the number of bacteria colonizing the liver and spleen, mice were infected by intraperitoneal injection with 2 × 10^5^ CFU of *S.* Typhimurium. At the specified days post-infection, approximately 0.2 g portions of livers and whole spleens were harvested and weighed and then homogenized in 1 mL of phosphate-buffered saline (PBS) using a tissue homogenizer. The homogenates were spread on LB agar plates at different dilution ratios to count the number of bacterial CFUs. Finally, the number of CFUs per gram of organ was calculated. To evaluate the effect of nitrate on *S.* Typhimurium virulence, drinking water containing 20 g/L of sodium nitrate was provided for mice from 3 days before i.p. infection to the end of the experiment. Meanwhile, the drinking water containing no sodium nitrate was used as a negative control.

### 4.4. Infection of Macrophages by S. Typhimurium

Different *S.* Typhimurium strains were cultured in LB medium at 37 °C for 12–15 h until the stationary phase, and they were then sub-cultured in fresh LB medium at an inoculation ratio of 1:100 for 12 h. Bacterial suspensions were diluted in complete DMEM (containing 4.00 mM glutamine, 4500 mg/L of glucose, and 10% FBS) at appropriate times. To reduce the surface charge of the bacteria, the diluted bacterial suspensions were shaken at 180 rpm at 37 °C for 30 min, and they were then gently added to the monolayer of RAW 264.7 cells at a multiplicity of infection of 10 (bacteria: macrophages). The culture plates were centrifuged at 1000 rpm for 5 min to synchronize the contact of the bacteria with macrophages and then incubated at 37 °C in an incubator-supplied 5% CO_2_ atmosphere for 40 min. The cells were gently washed twice with PBS to remove extracellular bacteria. Then, a complete medium containing high-concentration gentamicin (100 µg/mL) was added to the cell culture plates for 1.5 h to kill the residual extracellular bacteria, and from this point, the hours post-infection (hpi) were deemed as 0. Next, the cells were cultured in complete medium containing a low concentration of gentamicin (20 µg/mL). At 2 and 20 hpi, the infected cells were washed three times with PBS and lysed for 5 min with 0.1% Triton X-100. The cell lysates were serially diluted and spread onto LB agar plates. Bacterial CFUs were counted after incubation at 37 °C overnight. The relative intracellular replication was calculated as the CFU number at 20 h divided by that at 2 h.

### 4.5. Measurement of Nitrate and Nitrite Concentration

To measure the nitrate/nitrite concentration in the liver and spleen, approximately 0.2 g of liver and the whole spleen of the infected mice and uninfected mice (negative control) were harvested and weighed. They were then homogenized in 1 mL of PBS using a homogenizer and were next centrifuged at 10,000 rpm for 20 min. The supernatant liquids were filtered using ultrafiltration tubes that only allowed the discharge of small molecules, and during this time, the precipitates and enzymes that possibly influenced the measurement of nitrate/nitrite concentration were removed. The cell supernatants were subjected to the ultrafiltration process described above when the nitrate/nitrite concentration was measured. First, nitrate in the sample was converted to nitrite by nitrate reductases. Second, nitrite was converted to a deep-purple azo compound using Griess Reagent 1 (sulfanilamide) and Griess Reagent 2 (N-[1-Naphthyl] ethylenediamine). Finally, the total concentrations of nitrate and nitrite were calculated by measuring the absorbance of the deep purple azo chromophore at 540 nm (A_540_). To calculate the nitrite concentration of the same sample, only the two Griess reagents were added to the sample, and A_540_ was then measured. Different concentrations of nitrate standard solutions (from 0 to 40 µM) were prepared, and the nitrate reductases and Griess reagents were added to the standard solutions to test A_540_. The standard curve represents the linear relationship between the nitrate concentration (*x*-axis) and A_540_ (*y*-axis). The standard curve represents the linear relation of the nitrite concentration (*x*-axis), and A_540_ (*y*-axis) was constructed according to the methods described above. The intercept and slope of each standard curve were used to test the nitrate or nitrite concentrations of the samples according to Equations (1)–(3).
(1)$$\left[\mathrm{Nitrate}+\mathrm{nitrite}\right]\left(\mu M\right)=\left(\frac{A_{540}-y_{intercerpt}}{slope}\right)\left(\frac{200\;\mu L}{volume\;of\;sample}\right)\times \mathrm{dilution}$$
(2)$$\left[\mathrm{Nitrite}\right]\left(\mu M\right)=\left(\frac{A_{540}-y_{intercerpt}}{slope}\right)\left(\frac{200\;\mu L}{volume\;of\;sample}\right)\times \mathrm{dilution}$$
(3)[Nitrate](μM)=[Nitrate+nitrite]−[Nitrite]

### 4.6. Measurement of Intracellular pH of S. Typhimurium

*S.* Typhimurium was cultured in N-minimal medium for 12–16 h, and after this, it was sub-cultured in new medium. Six hours later, a portion of bacterial cultures were centrifuged at 5500 rpm to collect the supernatants, and then the supernatants were filtered by using a membrane filter with a pore size of 0.22 μm to obtain the bacteria-free medium. An amount of 20 μM of BCECF-AM was added to the N-minimal medium containing bacteria, and then it was shaken at 180 rpm under 30 °C. After 30 min, the BCECF-AM adhering to the bacterial surface was washed off 3 times with the bacteria-free medium described above, and then the fluorescence intensity at a 535 nm emission wavelength was measured using a microplate reader (Tecan, Männedorf, Switzerland) at excitation wavelengths of 488 (F488) and 440 nm (F440). Intracellular pH was calculated from the ratios of emission fluorescence intensities upon excitation at 488 and 440 nm (F488/F440) using the pH_in_ (intracellular pH value) standard curve that was prepared using the following method. First, 500 mL of N-minimal medium was aliquoted into five 100 mL portions, and the pH values were adjusted to 4, 5, 6, 7, and 8. The bacterial culture was centrifuged and collected, after which it was suspended in the same volume of N-minimal mediums with different pH, and the OD_600_ of the bacteria was adjusted to 0.3. Valinomycin (1 μM) and nigericin (1 μM) were then added to the bacterial suspensions. The bacterial suspensions were gently vortexed to achieve a balance between pH_in_ and extracellular pH. The bacteria were collected by centrifugation and suspended in the same N-minimal medium with different pH values. A 20 μM concentration of BCECF-AM was added to the bacterial suspensions at 30 °C for 20 min. The bacteria were collected by centrifugation, washed three times by using the N-minimal medium with the same pH value, and suspended again. The fluorescence intensity of the bacterial suspension was measured at an emission wavelength of 535 nm with excitation wavelengths of 488 nm and 440 nm. The fluorescence intensity (F) is the ratio of F488 to F440, and the standard curve was plotted with pH values as the *x*-axis value and lgF as the *y*-axis value [13,17].

### 4.7. RNA Preparation and qRT-PCR Analysis

For qRT-PCR analysis, bacteria were grown overnight in N-minimal medium (7.5 mM [NH4]_2_SO_4_, 1 mM KH_2_PO_4_, 5 mM KCl, 0.5 mM K_2_SO_4_, 10 μM MgCl_2_, 10 mM Tris-HCl (pH 7.5), 0.5% (*v*/*v*) glycerol, and 0.1% (*w*/*v*) casamino acids). To obtain the bacteria in stationary phase, the overnight-cultured bacteria were sub-cultured in fresh N-minimal medium (37 °C, 180 rpm) at a 1:100 inoculation ratio for 6 h. *S.* Typhimurium cells were collected by centrifugation at 10,000 rpm for 3 min and immediately frozen in liquid nitrogen. Total RNA was extracted using TRIzol reagent (Invitrogen, Waltham, MA, USA) according to the manufacturer’s instructions. Residual DNA in the total RNA samples was removed using a RNeasy Mini Kit (QIAGEN, Germantown, MD, USA). The final RNA concentration was determined by using a NanoDrop 2000 spectrophotometer (Thermo Fisher Scientific, Waltham, MA, USA).

qRT-PCR was performed using the QS5 Real-Time PCR system (Applied Biosystems, Waltham, MA, USA). The primers used for qRT-PCR analysis are listed in Appendix A. A total of 1.2 μg of RNA from each sample was reverse-transcribed into cDNA using the PrimeScript RT Master Mix (Perfect Real Time) (TaKaRa, Shiga, Japan). The qRT-PCR reaction mixture (30 μL) contained 15 μL of PowerUp™ SYBR™ Green Master Mix (Applied Biosystems, Waltham, MA, USA), 1 μL of cDNA (approximately 40 ng), 1 μL of forward primer, and 1 μL of reverse primer, each at a final concentration of 0.33 μM. The data were normalized to the 16S rRNA gene as a reference control. The target gene expression levels were calculated using the 2^−^^ΔΔCt^ method.

To quantify the gene expression of *S.* Typhimurium that resides in the mouse tissues, around 0.2 g of the *S.* Typhimurium-infected tissues were quickly harvested and homogenized by mixing with 0.5 mL of DNase/RNase-free water. The total RNA from the tissues and bacteria were extracted by using the TransZol Up Plus RNA Kit (TransGen Biotech, Beijing, China), which provided an effective solution that can simultaneously extract the RNA from both the animal tissues and bacteria. When performing qRT-PCR, a maximal cDNA amount should be used because the bacterial RNA takes up a small proportion of total RNA.

### 4.8. Expression of 6×His Tag Protein and Electrophoretic Mobility Shift Assay

The constructed pET28a-gene plasmid was transformed into *Escherichia coli* BL21 (DE3) cells. The DE3 bacteria harboring the pET28a-gene plasmid were cultured overnight in LB medium and sub-cultured in fresh LB medium at an inoculation rate of 1:100. When the OD600 was close to 0.6, 1 mM isopropylthio-β-galactoside (IPTG) was added to the LB medium to induce the expression of the 6 × His-tag protein. Sodium nitrate (30 mM) was added to the LB medium when expressing the NarL-6 × His protein to ensure that the protein was phosphorylated in response to nitrate. The protein was purified from the lysate supernatant from DE3 cells using a His-Tagged Protein Purification Kit (CoWin Biosciences, Taizhou, China). Protein concentration was determined according to the Bradford method, and the purified protein was stored at −80 °C.

EMSA was performed as previously described with some modifications. Briefly, 400 bp of DNA fragments consisting of the 300 bp upstream sequence and the first 100 bp of the target genes were amplified by PCR with high-fidelity enzymes and purified from gels, as well as the DNA fragments of 16S rDNA (negative control). The promoter fragments (50 ng) and different concentrations (0–1.6 μM) of the purified protein were incubated at room temperature for 30 min in a 20 μL solution containing EMSA binding buffer (20 mM Tris-HCl (pH 7.5), 50 mM KCl, 1 mM EDTA, 1 mM dithiothreitol, and 5% glycerol). Samples were then loaded onto a 6% polyacrylamide gel immersed in 0.5 × Tris-Borate-EDTA for electrophoresis. DNA fragments were stained with GelRed (Biotium, Fremont, CA, USA).

### 4.9. Statistical Analysis

Data were analyzed using GraphPad Prism (version 8.0.1; La Jolla, CA, USA). The data presented in each figure or table were obtained from three independent biological experiments (in vitro experiments) or a combination of two independent biological experiments (in vivo experiments). The Student’s *t*-test, log-rank (Mantel–Cox) test, or Mann–Whitney U test was used to analyze significant differences between the two groups according to the test requirements. Differences were considered significant at *p* < 0.05 (* *p* < 0.05; ** *p* < 0.01; *** *p* < 0.001; **** *p* < 0.001).

## 5. Conclusions

This work reports that nitrate levels in *S.* Typhimurium-infected macrophages and the liver and spleen of *S.* Typhimurium-infected mice are significantly increased, and nitrate utilization contributes to the intracellular replication and systemic infection of *S.* Typhimurium. The utilization of nitrate can increase the acidification of *S.* Typhimurium cytoplasm, which is a crucial signal to activate the expression of SPI-2 virulence genes. Nitrate utilization of *S.* Typhimurium is activated by both Fnr and NarX-NarL (see Figure 7 for a schematic of nitrate utilization in *S.* Typhimurium during systemic infection). These findings reveal a new mechanism related to the acidification of the *S.* Typhimurium cytoplasm and also provide new insights into the course of *Salmonella* systemic infection and the interactions between *Salmonella* and its host.

How the nitrate utilization acidifies the *S.* Typhimurium cytoplasm needs to be addressed in the future study, and whether the metabolic products of nitrate, such as nitrite or ammonium, contribute to the virulence of *S.* Typhimurium during systemic infection is worth exploring. *Salmonella. enterica* serovar Typhi (*S.* Typhi), the human-limited serovar, can cause lethal typhoid fever in humans via replicating inside human macrophages. The effect of host-derived nitrate on the virulence of *S.* Typhi can be assessed in the human macrophages and humanized mice if possible.

## Figures and Tables

**Figure 1 ijms-23-07220-f001:**
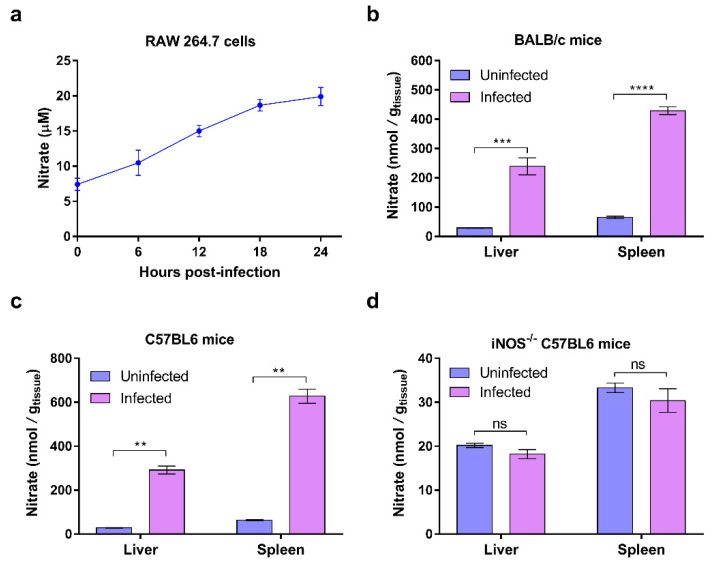
Nitrate levels are increased during systemic infection by *S.* Typhimurium. (**a**) Nitrate levels in RAW264.7 cells after 0, 6, 12, 18, and 24 h of infection with *S.* Typhimurium WT strain. (**b**,**c**) Nitrate levels in the livers and spleens of BALB/c mice (**b**) and C57BL/6 mice (**c**) that were infected or mock-infected with *S.* Typhimurium WT strain after 5 days post-infection. (**d**) Nitrate levels in the livers and spleens of iNOS-deficient C57BL/6 mice that were infected or mock-infected with *S.* Typhimurium WT strain after 5 days post-infection. (**a**–**d**) Data were generated from three independent experiments and are presented as the mean ± SD. *p*-values were determined using unpaired Student’s *t*-test (** *p* < 0.01; *** *p* < 0.001; **** *p* < 0.0001; ns, not significant).

**Figure 2 ijms-23-07220-f002:**
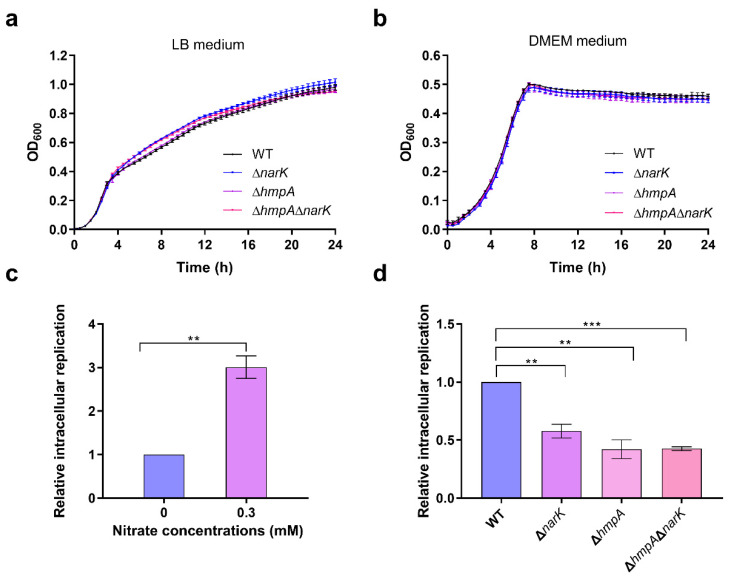
Nitrate utilization promotes *S.* Typhimurium replication in macrophages. (**a**,**b**) Growth curves for *S.* Typhimurium WT, Δ*narK*, Δ*hmpA*, and Δ*hmpA*Δ*narK* in LB medium (**a**) and DMEM medium (**b**). The absorbance of bacterial suspensions at 600 nm (OD6_00_) was measured regularly using a microplate reader over a 24 h time period. (**c**) Replication of *S.* Typhimurium WT in RAW264.7 cells in the presence or absence of 0.3 mM nitrate. The bacterial replication ability was determined according to the ratio of the number of intracellular bacteria at 20 h post-infection to the number of bacteria at 2 h post-infection. (**d**) Replication of *S.* Typhimurium WT, Δ*narK,* Δ*hmpA*, and Δ*hmpA*Δ*narK* in RAW264.7 cells. (**a**–**d**) Data were generated from three independent experiments and are presented as mean ± SD. *p*-values were determined using unpaired Student’s *t*-test (** *p* < 0.01; *** *p* < 0.001).

**Figure 3 ijms-23-07220-f003:**
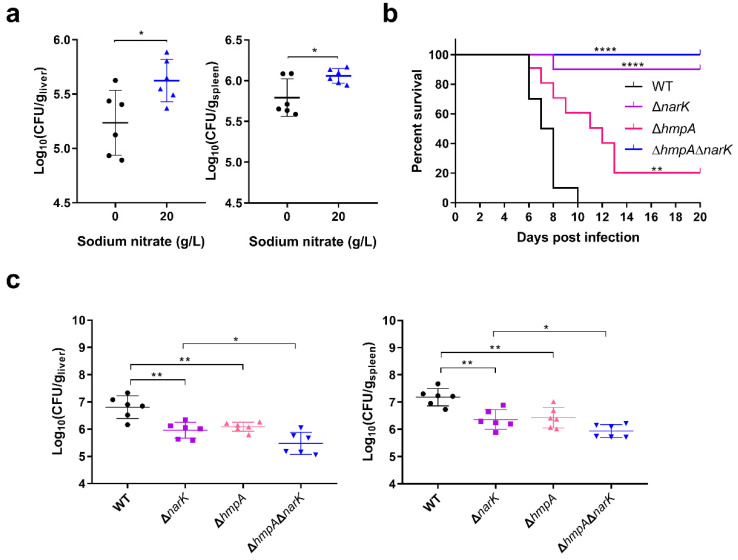
Nitrate utilization promotes *S.* Typhimurium systemic infection in the mice. (**a**) Drinking water containing 20 g/L of sodium nitrate was provided for mice from 3 days before i.p. infection to the end of the experiment. Meanwhile, the drinking water without the addition of sodium nitrate was used as control. Bacterial burdens of liver and spleen in infected mice were assessed at day 5 post-infection. n = 6 mice/group. (**b**) Survival curves for mice infected i.p. with the *S.* Typhimurium WT, Δ*narK*, Δ*hmpA*, or Δ*hmpA*Δ*narK,* n = 10 mice/group. (**c**) Liver and spleen bacterial burdens in mice infected with the *S.* Typhimurium WT, Δ*narK,* Δ*hmpA*, or Δ*hmpA*Δ*narK* at day 5 post-infection. n = 6 mice/group. (**a**–**c**) Data were combined from two independent experiments and are presented as mean ± SD. *p* values were determined using Mann–Whitney U test (**a**,**c**) or log-rank Mantel–Cox test (**b**) (* *p* < 0.05; ** *p* < 0.05; **** *p* < 0.0001).

**Figure 4 ijms-23-07220-f004:**
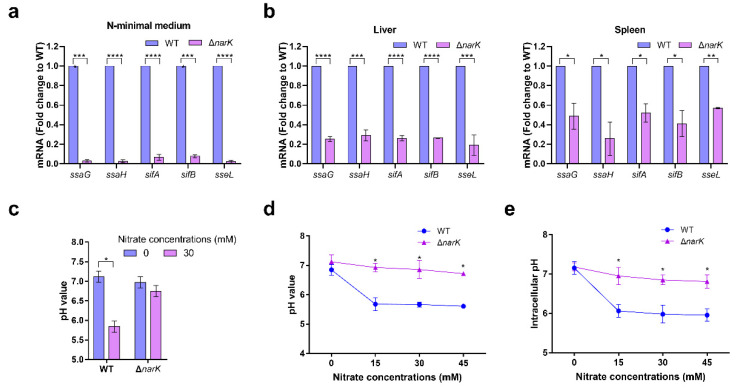
Nitrate utilization increases the intracellular acidification of *S.* Typhimurium to activate SPI-2 virulence gene expression. (**a**) qRT-PCR analysis of the mRNA levels of five SPI-2 genes in *S.* Typhimurium WT and the Δ*narK* mutant. Bacteria were grown in N-minimal medium supplemented with 30 mM nitrate for 6 h. (**b**) qRT-PCR analysis of the mRNA levels of five SPI-2 genes in *S.* Typhimurium WT and the Δ*narK* mutant that were extracted from mouse liver and spleen at day 3 post-infection. (**c**) *S.* Typhimurium WT and Δ*narK* mutant were cultured in N-minimal medium in the presence or absence of 30 mM nitrate for 6 h. Changes in the pH of the medium were assessed. (**d**) *S.* Typhimurium WT and Δ*narK* mutant were cultured in N-minimal medium in the presence of 0, 15, 30, or 45 mM nitrate for 6 h. Changes in the pH of the medium were assessed. (**e**) *S.* Typhimurium WT and Δ*narK* mutant were cultured in N-minimal medium in the presence of 0, 15, 30, or 45 mM nitrate for 6 h. Changes in the intrabacterial pH were assessed using a BCECF-AM fluorescent probe. (**a**–**e**) Data were generated from three independent experiments and are presented as mean ± SD. *p*-values were determined using unpaired Student’s *t*-test (* *p* < 0.05; ** *p* < 0.01; *** *p* < 0.001; **** *p* < 0.0001).

**Figure 5 ijms-23-07220-f005:**
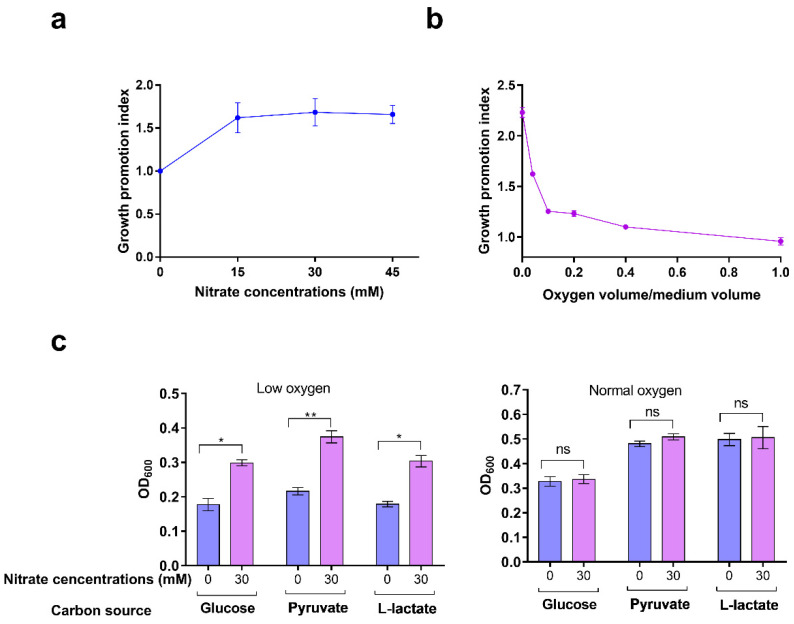
Nitrate utilization promotes *S.* Typhimurium growth under low-oxygen conditions. (**a**) *S.* Typhimurium WT strain was cultured in N-minimal medium in the presence of 0, 15, 30, or 45 mM nitrate for 6 h. The growth promotion index (GPI) of nitrate was calculated according to the OD_600_ of the bacterial culture supplemented with nitrate divided by that without nitrate. (**b**) *S.* Typhimurium WT strain was cultured in N-minimal medium in the presence or absence of 30 mM nitrate for 6 h with different oxygen levels. (**c**) *S.* Typhimurium WT strain was cultured in N-minimal medium under low-oxygen or normal-oxygen conditions with different carbon sources (35 mM glucose, 70 mM pyruvate, or 70 mM lactate) in the presence or absence of 30 mM nitrate. Absorbance of the bacterial suspension at 600 nm (OD_600_) was measured at 6 h. (**a**–**c**) Data were generated from three independent experiments and are presented as mean ± SD. *p*-values were determined using unpaired Student’s *t*-test (* *p* < 0.05; ** *p* < 0.01; ns, not significant).

**Figure 6 ijms-23-07220-f006:**
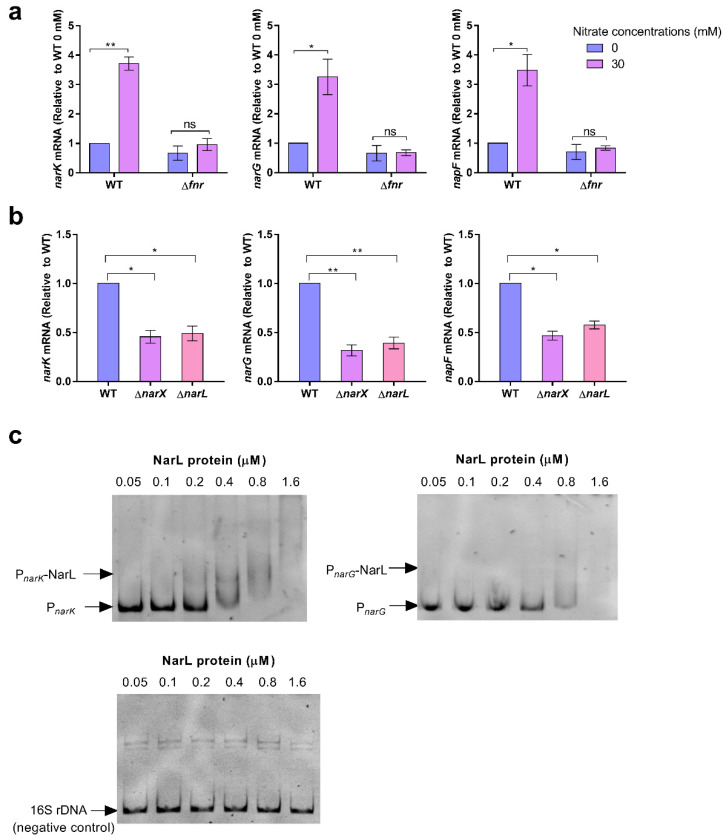
Nitrate utilization is activated by both Fnr and the nitrate-sensing two-component system NarX-NarL under low-oxygen conditions. (**a**) qRT-qPCR analysis of *narK*, *narG*, and *napF* mRNA levels in *S.* Typhimurium WT and the Δ*fnr* mutant. Bacteria were grown in N-minimal medium for 6 h prior to collection. (**b**) qRT-qPCR analysis of *narK*, *narG*, and *napF* mRNA levels in the *S.* Typhimurium WT, Δ*narX* mutant, and Δ*narL* mutant. Bacteria were grown in N-minimal medium for 6 h prior to collection. (**c**) EMSA of the *narK* and *narG* promoters with purified NarL protein. The 16S rDNA was used as a negative control. Images are representative of three independent experiments. (**a**,**b**) Data were generated from three independent experiments and are presented as mean ± SD. *p*-values were determined using unpaired Student’s *t*-test (* *p* < 0.05; ** *p* < 0.01; ns, not significant).

**Figure 7 ijms-23-07220-f007:**
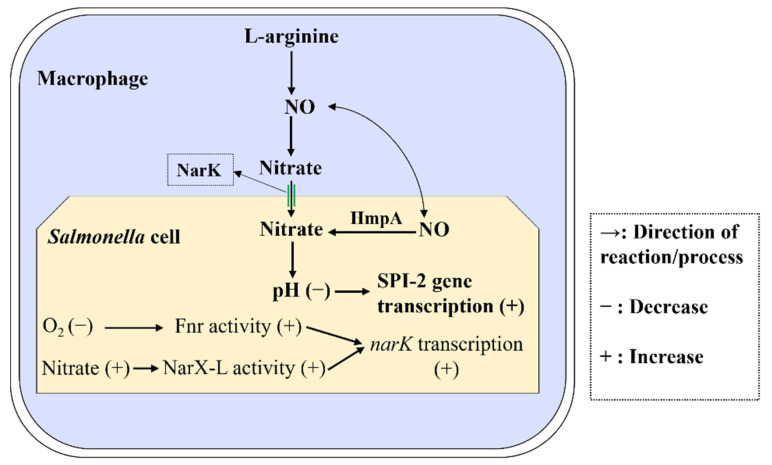
Schematic representation of nitrate utilization in *S.* Typhimurium during systemic infection.

**Table 1 ijms-23-07220-t001:** The influence of oxygen levels on the growth of *S.* Typhimurium.

Oxygen Volume/ Medium Volume	OD_600_		Growth Promotion Index (GPI)
	0 mM Nitrate	30 mM Nitrate	
0	0.123 ± 0.008	0.273 ± 0.011	2.230 ± 0.049
0.04	0.175 ± 0.009	0.284 ± 0.012	1.622 ± 0.013
0.1	0.253 ± 0.009	0.317 ± 0.009	1.254 ± 0.010
0.2	0.285 ± 0.009	0.351 ± 0.019	1.231 ± 0.028
0.4	0.352 ± 0.011	0.387 ± 0.004	1.100 ± 0.023
1	0.532 ± 0.007	0.510 ± 0.025	0.958 ± 0.036

## Data Availability

All data are presented within manuscript and Appendix A.

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
