# Peer review of "Nitrate Utilization Promotes Systemic Infection of Salmonella Typhimurium in Mice"

_ijms, 2022, doi:10.3390/ijms23137220_

Round 1
Reviewer 1 Report
In this manuscript, the authors tested important genes for nitrate utilization of Salmonella Typhimurium, and the effect of these mutations on bacterial growth in macrophages and in mice. The manuscript is well written with minor typos, and most of the experiments are well planned and executed. The results mostly support the conclusions, with a few exceptions below that either require a further discussion from the authors, or more experiments.
1. In Figure 2d, The growth defect of hmpA narK double mutant is similar to hmpA single mutant in macrophages. Since these are two different pathways of generating nitrate, why aren’t these two mutations additive, as shown in Figure 3?
2. In Figure 3a, unless the mice were deprived of nitrogen sources in the food, which is very unlikely, supplementation of NaNO3 in water is unlikely to affect the in vivo concentration of nitrate. To make such a claim that supplementation of NaNO3 in water affects the in vivo concentration of nitrate, one needs to measure nitrate concentration in the spleen and liver.
3. Figure 6c, the PnarK/narG-NarL band is almost invisible at 0.8 and 1.6 uM NarL protein concentration. Do you have an explanation for the lack of a clear promoter+NarL band?
Reviewer 2 Report
The study conducted by Li et al. is original with relevant contribution to the understanding of the roles of nitrate on S. Typhimurium systemic infection in Mice. I support its possible publication after appropriate minor modifications as outlined below.
The authors not used the line numbered template requested by the journal.
Abstract
“Here, we demonstrate that...” – please avoid the using of personal mode of verb forms throughout the manuscript. Please carefully revise this concern throughout the manuscript!
Introduction
“gram-negative” – capitalize Gram (sentence case)
The last paragraph “In this study, we investigated...” must completely rephrased. The authors must substantially shorten it, providing a clear definition of the study aim and without presenting some statements like “We discovered that” or “We observed that” – please rephrase them
Within the materials and methods section the authors must refer to the used negative and positive controls during molecular investigations in order to validate their results.
The authors must insert a conclusion section presenting the main conclusions, the study limitations as well as future perspectives in this research area
Reviewer 3 Report
This manuscript well describes the role of nitrate in Salmonella infection in mice. This study was to understand the effect of nitrate on the growth, replication, and virulence.
1. Then, overall schematic diagram of gene and protein expression should help to understand the nitrate utilization in Salmonella infection.
2. Data including the weights of spleen, liver, and mice should be also informative.
3. Is there any reason using both BALB/c mice and C57BL6 in this study?
4. The production of effector proteins, other Salmonella-infection associated gene expression, and tracking the number of infected Salmonella should be also provided.
